# Smartphone Use among Undergraduate STEM Students during COVID-19: An Opportunity for Higher Education?

**Javier Mella-Norambuena** [1], **Rubia Cobo-Rendon** [1], **Karla Lobos** [1], **Fabiola Sáez-Delgado** [2,*]
**and Alejandra Maldonado-Trapp** [3]

1   Laboratorio de Investigación e Innovación Educativa, IDECLAB, Dirección de Docencia,
    Universidad de Concepción, Concepción 4030, Chile; javimella@udec.cl (J.M.-N.);
    rubiacobo@udec.cl (R.C.-R.); karlalobos@udec.cl (K.L.)
2   Centro de Investigación en Educación y Desarrollo, Facultad de Educación,
    Universidad de Católica de la Santísima Concepción, Concepción 4030, Chile
3   Departamento de Física, Facultad Ciencias Físicas y Matemáticas, Universidad de Concepción,
    Concepción 4030, Chile; alemaldonado@udec.cl
*   Correspondence: fsaez@ucsc.cl

**Abstract:** Due to the COVID-19 pandemic, students worldwide have continued their education remotely. One of the challenges of this modality is that students need access to devices such as laptops and smartphones. Among these options, smartphones are the most accessible because of their lower price. This study analyzes the usage patterns of smartphone users of undergraduate Science, Technology, Engineering, and Math (STEM) students during the COVID-19 pandemic. This cross-sectional descriptive study included 365 students: 162 (44.4%) women and 203 (55.6%) men from a Chilean university. The results revealed that students often accessed the learning management system (LMS) with their computers rather than with their smartphones. Students were connected to the LMS for more hours on their computers than on their smartphones. However, they spent more hours simultaneously connected on their computers and smartphones than just on their computers. During the day, students accessed the LMS mainly from 13:00 to 1:00. The number of connections decreased from 1:00 to 8:00 and increased from 8:00 to 13:00. The LMS resource that students accessed the most using smartphones was discussion forums, while the one they accessed the least was wiki pages. We expect these results to motivate faculties to schedule their activities during the hours students tend to be online and promote discussion forums.

**Keywords:** smartphones and learning; mobile learning; university students; STEM; COVID-19

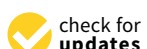

## 1. Introduction

A full year has passed since the World Health Organization (WHO) informed the world about the COVID-19 pandemic [1]. About 150 countries had to close their educational institutions by 25 March 2020, affecting more than 80% of the global student population [2]. Universities worldwide were closed for students in order to comply with physical distancing measures imposed by governments. This implied a sudden adaptation of educational processes from face-to-face and blended learning to emergency remote teaching. This complex scenario forced higher education institutions to face several obstacles and challenges [3,4], such as the virtualization of laboratory activities.

E-learning or electronic learning is a term comprising several types of electronically implemented learning, including blended learning (or B-learning) and online learning [5]. The latter type of electronic learning differs from others, such as technology-based learning [6,7], since online courses or educational programs are intentionally designed to be taught exclusively online. Prior to the COVID-19 pandemic, several educational institutions across the globe had begun implementing online or blended programs [8], but this health emergency forced all teachers and students in all countries to shift to online education without prior planning or training.

The pandemic caused an abrupt transition to online education; hence, the literature refers to this phenomenon as emergency remote teaching [9,10], with universities striving to adjust traditional teaching to a remote teaching format as quickly as possible. Emergency remote teaching (ERT) occurs in a context in which teachers use information and communication technology tools without necessarily possessing the training to teach students in a learning environment system (LMS). Although educational institutions globally considered remote teaching as a great opportunity to overcome the constraints caused by the pandemic [11], this modality was not carefully designed for taking advantage of and maximizing the possibilities offered by an online format [10]. Therefore, ERT is not recognized as online teaching but rather as an intermediate step toward it [12].

The COVID-19 pandemic has exposed emerging vulnerabilities in education systems around the world. It is now clear that human society needs flexible and resilient educational systems because it faces an unpredictable future. In underdeveloped countries, most students do not have access to the internet because of technical and monetary constraints. Other problems related to ERT include the absence of face-to-face interaction and a longer response time from the teachers to their students [13]. A study conducted with Chilean students indicated limited internet access resulting in poor student–teacher communication [14]. Another study with university teachers from India also reported that internet connection problems, a lack of training, and a lack of awareness were their main challenges. In addition, lower attendance to synchronic activities, a lack of personal contact, and a lack of interaction with students were found to be related to internet connection problems during virtual classes [15].

Recent meta-analytical research revealed that technological resources, teacher training, trust in the decisions institution authorities, access to internet, and student motivation play an important role in ERT. The authors conclude by proposing that teachers should use technology to improve learning, especially in these exception-al times [3]. The presence of these new challenges has an impact on students' academic adaptation processes and mental health. For example, a few studies have identified changes in students' habits due to confinement, which impact the duration and quality of sleep [16,17].

In ERT, both students and teachers must possess an electronic device connected to the internet to access online education. Desktop computers, laptops, tablets, and smartphones are the most popular devices. A flexibilization of education curricula is expected after the pandemic, not for shifting to exclusive online teaching but for integrating traditional teaching with online teaching through blended learning, taking advantage of the benefits provided by these methodologies [18,19]. The teaching methodology in which face-to-face classroom sessions are combined with online sessions is called blended learning—a relatively new model that began to be used and investigated only toward the end of the 1990s [20]. The main component of blended learning is the bimodal teaching method (face-to-face and online) that intends to integrate technologies, strategies, and pedagogical activities in a holistic, intentional, and efficient manner, thereby optimizing time by eliminating obstacles related to space, time, and resources [21].

Blended learning is based on the use of virtual tools that allow students to work remotely using an internet connection and devices for this purpose. Several countries have reported inequalities related to these aspects during the pandemic, which increase the knowledge gap between students and particularly affect those from economically underprivileged backgrounds [22]. A few universities have provided internet and computer internet scholarships and computer loans to their students, but this is not an option for all higher education institutions. Students who do not have access to these benefits must fund their own equipment to complete their online courses. Smartphones are often less expensive than personal computers and tablets. Therefore, the use of smartphones has allowed low socioeconomic status students to access remote education. Therefore, the implementation of learning resources and activities that are accessible through smartphones has become more important in the context of ERT.

*Smartphones for Higher Education*

Mobile learning, also known as M-learning [23], is a teaching and learning methodology that helps students conduct their educational activities in a simple, easy, and flexible manner, without temporal or spatial restrictions [24]. This methodology refers to the use of mobile devices for the implementation of learning resources and activities. Mobile learning allows students to participate in their courses from anywhere and at any time, and smartphones are among the most used devices in this context [25]. Mobile learning is considered a part of the online and blended learning models. It refers to the tools that shall be used to implement the teaching–learning process, while online and blended learning refer to the model used for the development of the teaching process [26,27]. M-learning is based on the use of smartphones, personal digital assistants, and tablets, which allow students to continue their education regardless of their location or time zone [27].

For students, the advantage of using these devices is the improvement in their level of interaction with teachers and classmates in real time during their academic training [28]. The student has immediate and permanent access to educational information anywhere or at any time [26], which means that they can learn without constraints of space and time [24]. Moreover, the use of smartphones for academic training is advantageous because of their relatively low cost as compared to computers [26,29].

Due to the versatility of a smartphone, its use in education has gained greater attention in educational research. Despite the widespread adoption of smartphones among young people, using them to facilitate learning in higher education is still a modern idea [29]. Therefore, research has been conducted to determine the different methods for adopting M-learning in higher education. Both teachers and students present positive levels of perception and acceptance toward M-learning [25,27,30,31]. There are also studies focusing on assessing the negative effects of smartphone use on the health of students, including ergonomics [32,33]; poor sleep quality [34,35]; mental health issues, such as stress, depression, and anxiety [36]; and psychological aspects, such as happiness and quality of life [37,38].

In the context of the COVID-19 pandemic, it is still not clear whether smartphones are an efficiently used resource by students. Therefore, it remains to be determined whether the use of smartphones by students improves learning or interferes with it. The evidence so far indicates that students facing the transition to ERT because of the pandemic have adopted B-learning, E-learning, and M-learning. They have had both positive and negative experiences during this adaptation process: they found teachers' actions to be limited in the implementation of these learning models, and they experienced the benefits regarding access to information anywhere or at any time [39].

Another study assessed the use of devices during ERT and found that 44.75% ($N = 400$) of students find it difficult to spend several hours of academic work in front of a mobile device or computer [40]. During the pandemic, students have been able to access lectures that are easily available to them and exchange a significant number of learning materials, but they have also reported challenges, such as visual fatigue, internet-related problems, the need for smartphones, and access to significant internet data that can be expensive for many [41]. As for the teaching methods, researchers have found that there is currently a lack of training and knowledge on the part of teachers about the different useful features of smartphones that can be used in teaching [29].

Alternatively, teachers are also concerned that smartphones can distract students from their studies. In the context of the pandemic, the use of smartphones as a resource for optimizing the learning of students is relevant. In this sense, assessing how students use smartphones in higher education is currently relevant.

Despite the effects of the pandemic on university education, it is important to determine how students are using smartphones as part of their academic training based on the current situation, especially in the case of Science, Technology, Engineering, and Math (STEM) students. These courses require a significant use of technology, and, as COVID-19 drastically reduced students' laboratory sessions and fieldwork, these were provided ex-

clusively via online teaching methodologies, such as simulations [42]. The knowledge of patterns of smartphone usage can influence the planning and success of online learning education methodologies and promote a quality transition to blended learning during the post-pandemic period in university education.

In this context, the objective of this study was to analyze the usage patterns of smartphone users of undergraduate STEM students during ERT because of the COVID-19 pandemic. Based on the literature and evidence described above, the following hypotheses are proposed:

**Hypothesis 1 (H1).** *Smartphone usage patterns of undergraduate STEM students for connecting to university LMS are similar to their usage patterns with their computer.*

**Hypothesis 2 (H2).** *There is no difference in the amount of time students spend active in the LMS using their smartphones and computers during the pandemic.*

**Hypothesis 3 (H3).** *Undergraduate STEM students were more active in the LMS at night than during the day.*

**Hypothesis 4 (H4).** *Using smartphones, undergraduate STEM students access diverse formats of educational resources, such as videos and infographics.*

**Hypothesis 4.1 (H4.1).** *Undergraduate STEM students accessed specific types of educational resources from their smartphones versus their personal computers.*

Our study proposes H1 and H2 based on the transition to the ERT due to the COVID-19 pandemic, with students having to attend their classes online [2]. H3 relates to the changes that have occurred as a result of the new online teaching models, the difficulties identified among students regarding their usage patterns [13], and the consequences for young people, particularly regarding sleep patterns, as a result of coping with these new challenges [16,17]. Hypotheses H4 and H4.1 are based on the empirical evidence on the advantages of using smartphones to access learning materials anywhere, at any time, and in any environment [26,43].

## 2. Materials and Methods

### 2.1. Participants

A total of 365 students from four different academic disciplines in a Chilean university participated in this study, namely 162 (44.4%) women and 203 (55.6%) men (mean age of 21.13, *SD* = 2.27): 65 (17.8%) from biology, 30 (8.2%) from physics, 232 (63.6%) from mathematics, and 38 (10.4%) from chemistry. All were students in the first two years of university education.

### 2.2. Instruments

Learning Analytics: Time, Connection per Used Device, and Access to Educational Resources.

Learning analytics from the institutional LMS Canvas were used to obtain information on smartphone use in basic science university students. The data provided by the Canvas LMS were analyzed, specifically from the Canvas Data Portal section [44]. This section has a log file called requests that tracks students' requests on the Canvas platform and their context, showing information such as the time of the request, the type of technological device used, the user who made the request, and the action taken. The URL (i.e., the URL used in the action) and user_agent (i.e., the field of the HTTP protocol that can be used to provide detailed information about the device used: computer or smartphone) were extracted from the requests to trace the action taken by students in each academic discipline [45].

In addition, the Canvas LMS allows teachers to organize the contents of their courses within modules, and each module contains resources of the following types:

1.  Quiz: corresponds to resources designed to perform formative or summative evaluations. It is generally presented to students at a predetermined time in the LMS and it can be answered directly on the platform or through attached documents. It is a resource that can be assigned to all students or to a particular group.
2.  Assignment: directed toward the execution of tasks, practical work, peer evaluation, etc. Like the Quiz, it can have an unlocking and expiration time and can be assigned to all students or to a particular group. Again, like the Quiz, it can be accessed and completed directly on the platform or through attached documents.
3.  Attachment: corresponds to documents that can be viewed directly on the platform (in mathematics, most attachments are ".m" files, which are Matlab or Octave codes) and files that can be uploaded to the platform and must be downloaded to be viewed (e.g., PDF, PPT, Word, compressed files, audio, images, or video).
4.  Discussion Topic: discussion forums where the teacher opens a conversation thread related to course content or social content. The students interact with each other in the forum, responding to their peers, creating content, or activating the 'like' option.
5.  WikiPage: pages produced in a rich text environment within the platform. The teachers can create material in the WikiPage containing definitions, exercises, videos, equations with LaTeX, videos, images, and infographics to complement. They are resources with great potential because, depending on the teacher's dedication, they can constitute complete and extremely useful pages for students.
6.  ExternalUrl: resources that redirect to external links. They are used to attach supplementary material.

Information about the interaction of students with the resources mentioned above was extracted from the URL and classified accordingly: Quiz, Assignment, Attachment, DiscussionTopic, WikiPage, or ExternalUrl. The device used to access these resources was extracted from the user_agent.

*2.3. Procedure*

The ethical standards for research with human subjects were considered when conducting the present study. Due to the use of electronic information generated since 2020 by the participation of students in the university's learning platforms, all students were presented with an informed consent form explaining the use of the digital fingerprint for the development of the research and the promotion of institutional educational policies that are beneficial to the entire educational community.

The Python programming language and regular expressions were applied for processing the learning analytics, using an algorithm for revising the information provided by the URL and user_agent to identify the type of resource with which the student interacts and the device from which the interaction was performed.

Different statistical procedures were implemented for the processed information according to the proposed hypotheses. In order to address H1, all connections made by students during the academic semester were considered according to the device used for access. The number of connections per device was plotted for each academic discipline. Normality tests were applied to analyze the distribution of connections. Results showed that distributions were non normal. In addition to the presence of outliers, the robust Yuen test was applied to examine significant differences and effect size in the use of each device for the different academic disciplines [46,47].

In order to address H2, a descriptive analysis of the hours students were connected to the Canvas LMS was performed. The types of connection were categorized into three groups: (1) computer session, (2) smartphone session, and (3) smartphone and computer session, in cases of simultaneous connections. Then, for each type of connection, it was assessed whether there were differences in the times according to the academic discipline.

For this purpose, normality and homoscedasticity tests were performed, and, in the absence of these, robust ANOVA tests were conducted [47].

In order to address H3, a histogram was first plotted showing students' connection–time patterns. Next, a robust ANOVA test was performed to determine whether a similar pattern existed for all academic disciplines because the data did not follow a normal distribution, the number of students per academic discipline was heterogeneous, and there were outliers [48].

In order to address H4, a descriptive analysis was performed regarding the educational resources available in the Canvas LMS. Next, the comparison was plotted in relation to the number of connections to each type of educational resource per device (smartphone or computer). All procedures described above were performed using the R version 4.0.5 and the RStudio IDE Version 1.3.959. The packages tidyverse, car, psych, WRS2, nortest, and reshape were used.

## 3. Results

The results are described according to the previously established hypotheses analyzing smartphone usage patterns of undergraduate STEM students during the ERT caused by the COVID-19 pandemic. The results are presented as the number of connections according to each device (smartphone or computer) to assess smartphone use patterns during the period selected for the analysis (H1). Likewise, the students' connection hours are described in general and by academic discipline for assessing the connection hours according to each device (H2) and the connection patterns throughout the day (H3). Finally, the types of educational resources found in the Canvas LMS platform for each academic discipline are described. The types of educational resources that students access through smartphones (H4) are analyzed.

### 3.1. Students' Connections per Device

H1 intends to determine whether the students' connection pattern using a smartphone is like their connection pattern using a computer. In order to address this, the connections made from 13 April to 30 August of 2020 to the Canvas LMS by students of four academic disciplines were analyzed. A typical academic year in Chile starts in March and finishes in January. However, due to COVID-19, the first semester of 2020 started in April and finished in August.

The results indicate that students made a total of 517,314 connections: 497,618 (96.2%) using a computer and 19,696 (3.8%) using a smartphone. Regarding the types of connections according to each academic discipline, there were 95,601, 6278, 351,376, and 64,059 connections for biology, physics, mathematics, and chemistry, respectively. Figure 1 shows the number of connections per device for each academic discipline.

Considering the total number of connections, physics students used smartphones the most, mathematics students the least. To analyze the differences between the number of connections per device in each academic discipline, four robust Yuen tests were performed for paired samples. On average, biology students connected 88.91 ($SD = 69.98$) times from computers and 1360 ($SD = 774.89$) times from smartphones. The robust Yuen test for related samples was significant in this case ($t(40) = 12.0176$, $p < 0.001$, $ES = 0.91$), indicating that most connections were made using computers.

Regarding physics, on average, students connected 49.18 ($SD = 66.22$) times via smartphones and 148.52 ($SD = 87.75$) times via computers. When comparing smartphone use and computer use, the robust Yuen test for related samples was significant ($t(20) = 8.374$, $p < 0.001$, $ES = 0.9$), in favor of the use of computers.

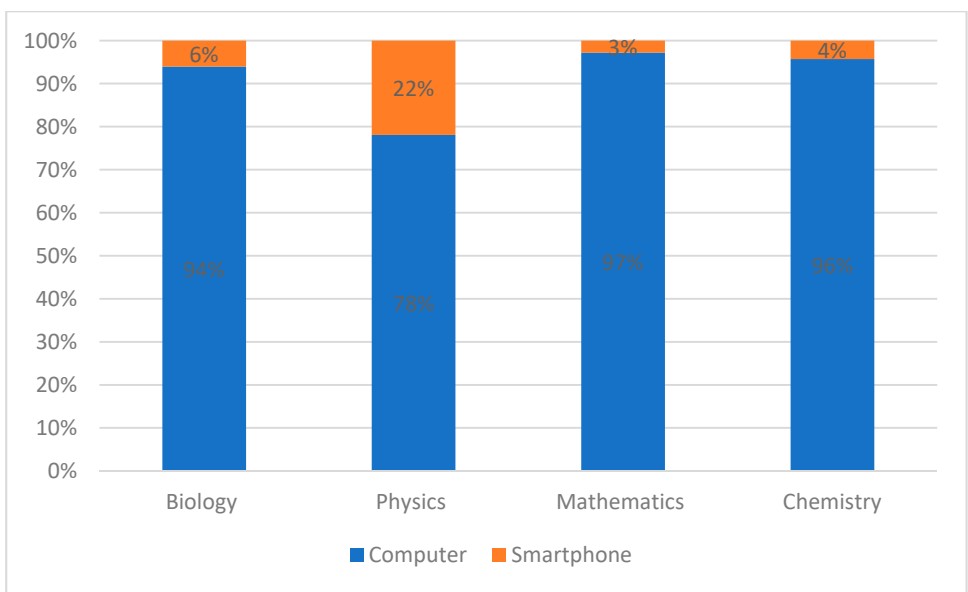

**Figure 1.** Connections (in %) of participating students according to each device.

As for chemistry, on average, students connected 83.12 (*SD* = 163.33) times from smartphones and 1495.51 (*SD* = 1009.82) times from computers. The robust Yuen test for related samples was significant (*t*(24) = 11.8756, *p* < 0.001, *ES* = 0.93), in favor of the use of computers.

For mathematics, on average, students connected 44.18 (*SD* = 62.33) times from smartphones and 1371.80 (*SD* = 938.78) times from computers. The robust Yuen test for related samples was significant (*t*(150) = 27.8707, *p* < 0.001, *ES* = 0.98), in favor of the use of computers. These findings reveal that the participating students, in general, made fewer connections using smartphones, and physics students most often used this device to access the contents of their course. These connection patterns demonstrate that there are differences in students' connection patterns according to device, which differs from the proposed hypothesis.

### 3.2. Connection Times per Device

H2 intended to confirm whether students' connection times with smartphones and computers were the same during the pandemic because of the ERT. On average, the participating students were connected 79.65 h to the LMS: 69.89 h exclusively from computers, 1.64 h exclusively from smartphones, and 8.11 h using both devices.

The analysis considered the total hours those students accessed the Canvas LMS with three types of connection: (a) only smartphone, (b) only computer, and (c) both devices.

Regarding the connection times per device for each academic discipline, on average, physics students spent more time connected to the classroom, dedicating 98.16 h. As for access per device, biology students stayed the longest in the classroom with this device, with 4.03 h. Similarly, mathematics students used smartphones the least, with a mean of 0.75 h. Table 1 shows measures of central tendency for connection times according to device and academic discipline.

**Table 1.** Measures of central tendency for connection times according to device and STEM discipline.

| Time per Device | Biology | | Physics | | Mathematics | | Chemistry | |
|---|---|---|---|---|---|---|---|---|
| | *M* | *SD* | *M* | *SD* | *M* | *SD* | *M* | *SD* |
| Smartphone | 4.03 | 5.87 | 2.05 | 5.22 | 0.75 | 1.40 | 2.74 | 6.47 |
| Computer | 70.26 | 39.37 | 87.58 | 63.95 | 66.91 | 49.04 | 75.08 | 45.27 |
| Computer and smartphone (simultaneous) | 13.23 | 12.14 | 8.53 | 11.62 | 7.26 | 10.55 | 4.18 | 6.15 |
| Total | 87.52 | 39.95 | 98.16 | 69.60 | 74.92 | 51.73 | 81.99 | 44.14 |

Regarding the comparison of connection times from smartphones by academic discipline, the Kolmogorov–Smirnov test results were first analyzed with the Lilliefors modification, which indicated that the data on the academic disciplines follow a normal distribution with respect to smartphone use time ($p < 0.001$). Furthermore, the Levene's test ($p < 0.001$) shows lack of homoscedasticity. A robust ANOVA test was performed because of the lack of normality, lack of homoscedasticity, presence of outliers, and heterogeneity in the amount of data per group, and it was statistically significant ($F(3,39.86) = 7.3532$, $p < 0.001$). The post hoc test shows that there are significant differences in smartphone connection times between biology and physics students ($p = 0.0017$) and between biology and mathematics students ($p < 0.001$).

As for the comparison of connection times using computers according to each academic discipline, the Kolmogorov–Smirnov test with the Lilliefors modification indicates that the data follow a normal distribution only for biology ($p = 0.399$). Moreover, Levene's test ($p = 0.439$) does not show lack of homoscedasticity. A robust ANOVA test was performed owing to the lack of a normal distribution of the data for three of the four groups, the presence of outliers, and the heterogeneity in the amount of data per group, and it was not significant ($F(3,45.96) = 1.908$, $p = 0.1439$). Therefore, it can be assumed that there are no differences between the connection time using a computer according to the academic discipline.

Finally, regarding the comparison of connection times from smartphones and computers simultaneously for each academic discipline, the Kolmogorov–Smirnov test with the Lilliefors modification indicates that the entire data does not follow a normal distribution ($p < 0.01$). Moreover, the Levene's test ($p = 0.054$) does not show lack of homoscedasticity. A robust ANOVA test was performed because of the lack of normality for the four groups, presence of outliers, and heterogeneity in the amount of data per group, and it was significant ($F(3,46.3) = 7.1238$, $p < 0.001$). Therefore, there are differences between the number of hours of simultaneous connection to the platform using computers and smartphones per academic discipline. The post hoc test shows that there are significant differences in the duration of connection with the simultaneous use of smartphones and computers between biology and mathematics ($p < 0.001$) and between biology and chemistry ($p < 0.001$), and biology students were connected for more time simultaneously.

### 3.3. Students' Connection Hours to the Canvas LMS

In order to determine whether students connected to the LMS more at night during the pandemic owing to the ERT (H3), their connection hours were depicted as a function of the analyzed period. Figure 2 shows the number of connections per hour of the day for all students, and the greatest number of connections occurs after 3:00 p.m. Students remained connected to the LMS platform at night and in the early hours of the morning.

A robust ANOVA test, which was performed to determine whether this connection pattern occurs in all academic disciplines, was not significant ($F(3680.72) = 0.1517$, $p = 0.92859$). Thus, this pattern is the same for students of all four academic disciplines.

### 3.4. Types of Educational Materials Accessed per Device

The materials provided on the Canvas LMS for the different academic disciplines were analyzed to address the hypothesis related to students' access to different types of educational materials from smartphones (H4 and H4.1). According to the type of materials provided in the analyzed academic disciplines, 310 correspond to WikiPage (47.98%), 126 to Attachment (19.50%), 115 to Quiz (17.80%), 58 to DiscussionTopic (8.97), 33 to Assignment (5.10%), and only four to ExternalUrl (0.61%). Moreover, teachers of the four academic disciplines entered a total of 646 resources in the LMS related to their virtual classrooms, including 138 (21.36%) for biology, 144 (22.29%) for chemistry, 135 (20.89%) for physics, and 229 (35.44%) for mathematics—this last discipline showed the highest percentage of educational materials provided to its students. Figure 3 shows the number of student connections according to the type of device and educational resource.

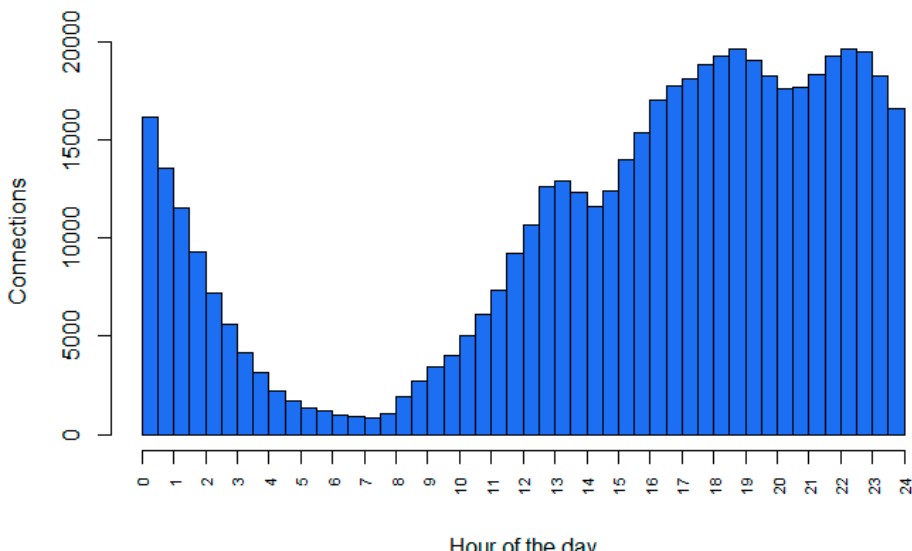

**Figure 2.** Number of students' connections to the Canvas LMS per hour of the day.

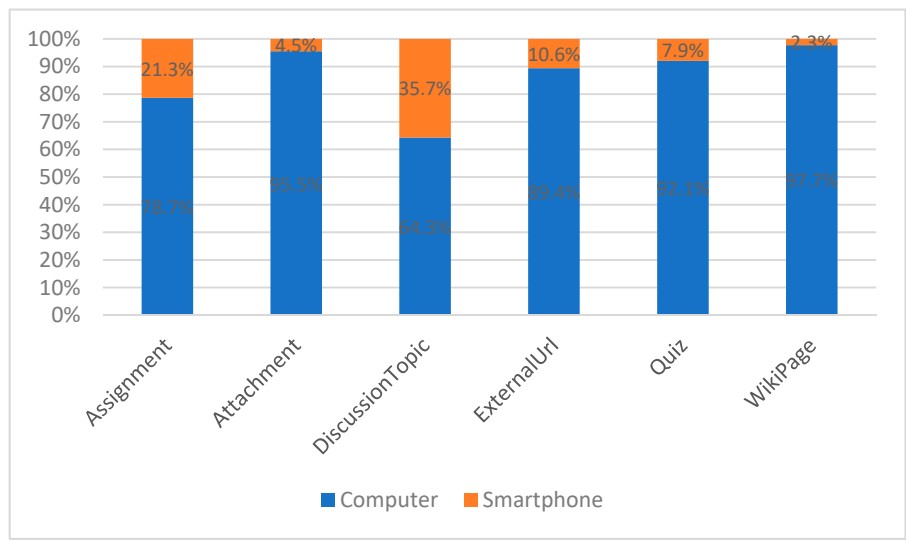

**Figure 3.** Access to different types of resources according to device.

DiscussionTopic was the resource accessed by students in the most balanced manner: 35.7% using smartphones and 64.3% using computers. WikiPage, which was the educational resource least accessed via smartphones (2.3%), was the predominant educational resource accessed in the LMS of the analyzed disciplines (47.98%).

## 4. Discussion

Smartphones bring us together even when unusual scenarios such as the COVID-19 pandemic force us to maintain physical distance. As life must go on, university students should continue their educational training. Maintaining that physical distance should not be complicated if technological resources are used to meet academic goals. Smartphone applications (apps) can maximize learning possibilities when used by university students. This is true if students not only have apps for entertainment but also have educational apps installed [48]. Therefore, the objective of this study was to analyze the usage patterns of smartphone users of undergraduate STEM students. The findings are discussed, considering the proposed hypotheses.

### 4.1. Students' Connections per Device

As a result of the transition to ERT caused by the pandemic, students have had to continue their studies using online education models [2]. Regarding the hypothesis concerning whether the students' connection patterns via smartphones are similar to those via computers, the findings indicate a higher percentage of computer usage than smartphone usage. Analyzing the users' activity patterns in the LMS revealed that biology students connected more often exclusively using smartphones and using them simultaneously with computers than with computers only. This finding contradicts another study conducted in the context of COVID-19, which reports that students ($n$ = 178; 57.98%) in other universities preferred using smartphones for online learning.

The article highlights the utility of using smartphones for emergency remote teaching in the COVID-19 context in the mentioned study. In the article, the authors detailed how teachers used smartphones for different learning processes. For example, teachers can obtain instant feedback from the students by implementing surveys or games via Kahoot. Moreover, students can access electronic books or documents, perform online searches, or practice topics from diverse areas [49]. In Legault et al. (2020), the authors showed how teachers' actions could support and increase smartphone usage among students.

University students are supposedly prepared for M-learning due to their familiarity with this technology and experience of accessing these types of resources. However, for M-learning to prosper in higher education, institutional and educational actions are necessary to facilitate the implementation of this type of online teaching, particularly the preparation of fluent experiences that encourage students to adopt M-learning [25]. Therefore, it is essential to assess aspects such as familiarity, practicality, and ease of smartphone use, which positively influence the use of these devices to perform academic activities [23,50,51].

### 4.2. Students' Connection Times to Learning Platforms

Students' active time using smartphones and computers is addressed in H2. We found that undergraduate STEM students were active in the LMS using their computers for a longer time than when using their mobile devices. Notably, the connection peaks with computers coincide with the connection peaks with smartphones. Students perform certain activities with both devices, indicating the simultaneous use of these resources during online education experiences.

The inclusion of smartphones and digital devices has quickly increased globally, leading to new approaches to learning through mobile technologies [52]. This finding demonstrates the implementation of mobile technology in higher education. Existing evidence reveals that students recognize that smartphone usage promotes critical thinking, innovative thinking, problem solving, communication, and teamwork skills [51].

### 4.3. Students' Connection Patterns to the Canvas LMS

Based on these results, although there are studies that consider smartphones harmful for students, the literature also supports that these technological devices can be an efficient tool to increase and improve learning if used in a responsible, well-informed, and timely manner [49]. In addition, smartphones have been cited as a valuable tool for remote learning during the COVID-19 pandemic, supporting practical lessons in education [50]. As most students own smartphones, this small tool has been used for reaching students and achieving a significant learning impact.

Student–teacher interaction is a crucial element to the success of the teaching and learning process [29]. The usage patterns of students' connection to the LMS show that students interacted more often during non-working hours, from evening to early night (5 p.m. to 1 a.m.), than during typical class hours (from 8 a.m. to 7 p.m.). This pattern connection could be explained by the lack of internet access during typical class hours and not having an optimal working space at home due to confinement [14]. The connection patterns suggest that students use smartphones autonomously without the immediate

support of teachers. The asynchronous interactions between students and teachers could be decreasing the advantages and benefits of smartphones [28].

On the other side, the pattern of smartphone usage could negatively affect students' health. For example, research performed in the COVID-19 context shows that higher education students who sleep less than 6 h per night tend to develop depression and stress symptoms [16]. These results agree with other studies in which students expressed having a decrease in their sleep quality [51,52]. Therefore, to obtain significant benefits from the usage of smartphones in education, teachers must implement pedagogical decisions that promote efficient and healthy interactions between students and the LMS. An example of the latter could be blocking access to activities during the night.

### 4.4. Types of Educational Materials Accessed per Device

Regarding students' access to educational materials, it was hypothesized that they would access various types of materials using smartphones, and this is the resource used for online classes. The resources most often accessed by students using smartphones include DiscussionTopic, better known as discussion forums—a space where the teacher opens a conversation thread related to course content or social content—and the tasks or assignments provided in the resource titled Assignment.

This finding highlights the advantages of using smartphones to access various learning materials anywhere, anytime, and in any environment [26,43]. However, these materials must be designed considering accessibility via smartphones. Therefore, universities should promote the development of teachers' skills and knowledge regarding the characteristics and benefits of smartphones in online education. It is also important that universities and teachers create appropriate educational content and web resources for both computer and smartphone screens [29,53].

It is important to consider that this study has limitations. This cross-sectional study analyzed students' behavior from one Chilean university taking courses on four disciplines during a particular semester, i.e., the first COVID-19 semester. Although student data correspond to one of the largest universities in Chile, with more than 20,000 students, the study only considers 365 students in their first or second year of college. A total of 17.8% of subjects took courses from biology, 8.2% from physics, 63.6% from mathematics, and 10.4% from chemistry. From the percentages, it is possible to note that most of the students were taking courses in mathematics, and the minority were taking courses in physics. This non-homogeneous distribution could have affected the study results, and, therefore, they need to be interpreted carefully.

This study only presents descriptive information regarding the students' behavior; thus, it is important that future studies further examine these types of behavior patterns using mixed or qualitative methodologies to find out the reasons and motives that lead students to have such behaviors. Finally, it would be relevant to further examine the use of M-learning in terms of other topics of interest in university education, such as academic performance, perceived usefulness, and perceived ease of use of smartphones in university education, especially in case of students from other STEM areas who are significantly exposed to technological resources [54,55].

Practical implications from this study are related to the need to educate teachers on the benefits of smartphone use for education and its advantages over using personal computers. It is necessary to create awareness among teachers regarding the assets that smartphone usage adds to online and blended learning modalities, especially since it is expected that blended learning will become more popular once the COVID-19 pandemic is over [56,57].

The lack of knowledge about smartphones' opportunities limits teachers from utilizing them for implementing active learning techniques. These techniques improve the learning process and increase the cognitive levels that students can reach [58]. An example of this is using the electronic immediate feedback technique through applications such as Kahoot, Mentimeter, Google forms, or Microsoft forms. This feedback allows teachers to find out

what their students are thinking in an easy and fast way. With this information, teachers can create discussion questions to motivate students to reflect on specific topics to reach the course's learning goals.

The more frequent access to discussion forums than to other resources using smartphones could be explained by the fact that teachers usually do not develop learning resources and activities considering the smartphone mobile interface. This creates different conflicts for students to interact with the material; for example, they can have visualization or download issues.

The strength of this study lies in the use of learning analytics for the analysis of students' behavior during online courses. Due to the COVID-19 pandemic, universities have adopted the LMS to facilitate teaching and learning processes. One of the benefits of this type of teaching resource is that it enables access to the digital fingerprint of students' behavior related to course materials and content. This is beneficial for teachers to provide information on shared resources, among other processes existing in teaching and learning scenarios.

## 5. Conclusions

From the study results, the authors conclude the following:

Smartphones are still not being used as a teaching and learning tool. Students scarcely use them for academic purposes, and, when they do use them, they do it during the night. It is necessary that faculty training instructs teachers on the planning and implementation of learning resources and activities with smartphones in mind during the design. The latter's objective is to increase student participation in synchronic and asynchronous online activities.

The higher use of smartphones by students in biology courses shows that it is possible to obtain higher participation through smartphones. However, in this context, it is essential to consider the role of teachers in promoting the use of the device.

The COVID-19 pandemic has given educational researchers a unique opportunity to study the use and utility of smartphones in the context of teaching and learning processes. Mobile learning via smartphones allows access to learning in different contexts; this has made it possible to move toward ubiquitous learning based on the strengths of mobile devices. This new context has open, diverse research lines that allow discussion and reflection on devices such as smartphones for teaching and learning. Since smartphones are already part of society, especially among young people, the device can be helpful for teachers and higher education institutions.

**Author Contributions:** The contributions of the authors of this paper were as follows: Conceptualisation, J.M.-N., R.C.-R. and K.L.; methodology, J.M.-N., R.C.-R. and K.L.; formal analysis, J.M.-N.; research R.C.-R., F.S.-D., A.M.-T. and K.L.; resources, J.M.-N., R.C.-R., F.S.-D., A.M.-T. and K.L.; data curation, J.M.-N. and R.C.-R.; writing-preparing the original draft, R.C.-R., F.S.-D., A.M.-T. and K.L.; writing-revising and editing, J.M.-N., R.C.-R., F.S.-D., A.M.-T. and K.L.; visualisation, J.M.-N.; monitoring, R.C.-R., A.M.-T. and K.L.; project management, R.C.-R., F.S.-D. and K.L.; fundraising, R.C.-R., A.M.-T. and K.L. All authors have read and agreed to the published version of the manuscript.

**Funding:** This research was funded by Unidad de Fortalecimiento Institucional of the Ministerio de Educación Chile, project InES 2018 UCO1808 Laboratorio de Innovación educativa basada en investigación para el fortalecimiento de los aprendizajes de ciencias básicas en la Universidad de Concepción.

**Institutional Review Board Statement:** The study was conducted according to the guidelines of the Declaration of Helsinki, and approved by the Institutional Ethics Committee of University of Concepción (protocol code CEBBE-656-2020, date of approval April 2020).

**Informed Consent Statement:** Informed consent was obtained from all subjects involved in the study.

**Data Availability Statement:** Further inquiries can be directed to the corresponding author/s.

**Conflicts of Interest:** The authors declare no conflict of interest.

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
