# Peer review of "Smartphone Use among Undergraduate STEM Students during COVID-19: An Opportunity for Higher Education?"

_education, doi:10.3390/educsci11080417_

Round 1

Reviewer 1 Report

  1. This is a meaningful study as this manuscript contributes to smartphone use among undergraduate STEM students during COVID-19.
  2. The title, abstract, and introduction are appropriate.
  3. The full-length article is well constructed.
  4. The analysis of materials and methods is well done.
  5. The value of this study as a reference for higher education is current.
  6. From the perspective of the educators, one suggestion is to include a description in the discussion and conclusion section about how the findings of this study can be applied in all fields of higher education.

Author Response

Dear,

We are writing in relation to manuscript ID: education-1304042, entitled " Smartphone use Among Undergraduate STEM Students During COVID-19: An Opportunity for Higher Education?". Authors: Javier Mella-Norambuena, Rubia Cobo-Rendon, Karla Lobos, Fabiola Sáez-Delgado*, Alejandra Maldonado-Trapp. 
We would like to thank for their comments, which we believe have greatly improved the quality of our article. We have modified the text accordingly and respond to each of the issues raised below. 

Table 1. Reviewers’ comments and revisions with corresponding improvement

Reviewer 1

Reviewer's comment

Authors' response

This is a meaningful study as this manuscript contributes to smartphone use among undergraduate STEM students during COVID-19.

No modifications required

The title, abstract, and introduction are appropriate.

No modifications required

The analysis of materials and methods is well done.

No modifications required

The value of this study as a reference for higher education is current.

No modifications required

The full-length article is well constructed.

No modifications required

From the perspective of the educators, one suggestion is to include a description in the discussion and conclusion section about how the findings of this study can be applied in all fields of higher education.

The suggestion is highly appreciated.; a paragraph was incorporated in the discussion and conclusion on how the findings of this study can be applied in all fields of higher education.

Reviewer 2 Report

This article addresses the use of communication devices for STEAM students from a Chilean university to access educational resources during COVID-19, specifically through an LMS. The introduction is well written and mentions a good set of current references. The study aims “to analyze usage patterns of smartphone users” during COVID-19 time, and also the usage patterns of computer and smartphone-and-computer simultaneously. Hypotheses are established that result from the introduction and are in accordance with the objective of the study. However, these hypotheses seem unambitious, as they relate mainly to the frequency of use of devices in a set of "specific types of educational resources".

The study involved students from four areas, namely biology, physics, mathematics, and chemistry. Although the total number of students is relevant, there is a significant difference between those areas. In fact, the study includes 63.3% of students from mathematics, and much less proportions from all the other areas. However this is described in section 2.1, it is not addressed or discussed in the results section and subsequent sections.

The methods and ethics are well described, and students gave an informed consent to the use of their usage data.

The results are presented in line with the hypotheses. I suggest the author to change the colors used in the graphic of Figure 2: the smartphone data is in blue in Figures 1 and 3, and in orange in Figure 2; the opposite happens with respect to computer data.

I found the Discussion and subsequent sections should be more elaborated. I general, they fall short on arguments, maybe because the hypotheses are just based on the usage patterns, in a descriptive way. More specifically, my main issues are:

  • In section 4.1, the author compares the results of biology students with the results from a different study, in India, arguing that both results are contradictory. This claiming must be demonstrated and justified. In fact, even if the numbers point to different directions, they respect to different contexts, which is not acknowledge.
  • Please show more clearly how the section 4.3 relates to the previously presented Results.
  • I suggest a new section from line 445, where the merits of this study could be more elaborated. The disproportion of mathematics students should be included.
  • The fact that the data is just from one University and was collected in the COVID-19 period should be addressed.
  • The “practical implications” (line 456) are not related to the Results of the study, which does not address subjects such as smartphone support of “active methodologies”.
  • The Conclusions presented are too general and seem to be more as a short version of results rather than the conclusions of the study. I suggest rewriting the Conclusions section;

Finally, the subtitle of the paper puts a question that is not discussed, and I am not convinced that this could be discussed under the subject of this paper.

I wish the authors all the best for this work and their future work

Author Response

Dear, 

We are writing in relation to manuscript ID: education-1304042, entitled " Smartphone use Among Undergraduate STEM Students During COVID-19: An Opportunity for Higher Education?". Authors: Javier Mella-Norambuena, Rubia Cobo-Rendon, Karla Lobos, Fabiola Sáez-Delgado*, Alejandra Maldonado-Trapp.  

We would like to thank for their comments, which we believe have greatly improved the quality of our article. We have modified the text accordingly and respond to each of the issues raised below.  

Table 1. Reviewers’ comments and revisions with corresponding improvement 

Reviewer 2 

Reviewer's comment 

Authors' response 

This article addresses the use of communication devices for STEAM students from a Chilean university to access educational resources during COVID-19, specifically through an LMS. The introduction is well written and mentions a good set of current references.  

No modifications required 

The study aims “to analyze usage patterns of smartphone users” during COVID-19 time, and also the usage patterns of computer and smartphone-and-computer simultaneously. Hypotheses are established that result from the introduction and are in accordance with the objective of the study. However, these hypotheses seem unambitious, as they relate mainly to the frequency of use of devices in a set of "specific types of educational resources". 

Regularly, the university where the study was carried out focuses mostly in the implementation of face-to-face courses. The online modality was implemented due to the pandemic. Therefore, there is no previous data of the contributions of learning analytics to understand the teaching and learning processes of students. Specifically, there is no previous data that characterizes the use of smartphones for academic processes.  

Since online teaching was implemented, there has been an important transition in the framwork of using learning analytics. The perspective moved from being mostly remedial (with the use of classic indicators such as attendance and performance), to a preventive one that addresses the performance of students before failure occurs (with indicators of the interaction of the students with the resources and activities of the virtual classrooms) . In addition, in Latin America, research on learning analytics  regarding the understanding of teaching-learning processes is scarce, even more so when focusing on the use of smartphones. Therefore, the hypotheses established in this study and the respective findings presented in this article are a huge contribution to the educational community.  

See:  

Ifenthaler, D., & Yau, J. (2020). Utilising learning analytics to support study success in higher education: a systematic review. Educational Technology Research and Development, 68(4), 1961-1990. https://doi.org/10.1007/s11423-020-09788-z 

Rojas-Castro, P. (2017). Learning Analytics: una revisión de la literatura. Educación y Educadores, 20(1), 106-128. http://dx.doi.org/10.5294/edu.2017.20.1.6 

The study involved students from four areas, namely biology, physics, mathematics, and chemistry. Although the total number of students is relevant, there is a significant difference between those areas. In fact, the study includes 63.3% of students from mathematics, and much less proportions from all the other areas. However this is described in section 2.1, it is not addressed or discussed in the results section and subsequent sections. 

Reviewer suggestion is appreciated. 

Information was incorporated into study limitations on lines 471-476. 

The methods and ethics are well described, and students gave an informed consent to the use of their usage data. 

No modifications required 

The results are presented in line with the hypotheses. I suggest the author to change the colors used in the graphic of Figure 2: the smartphone data is in blue in Figures 1 and 3, and in orange in Figure 2; the opposite happens with respect to computer data. 

The suggestion is appreciated. 

Graphics colors have been changed. The orange smartphone was approved in the graphics and the computer in blue. 

I found the Discussion and subsequent sections should be more elaborated. I general, they fall short on arguments, maybe because the hypotheses are just based on the usage patterns, in a descriptive way. More specifically, my main issues are: 

In section 4.1, the author compares the results of biology students with the results from a different study, in India, arguing that both results are contradictory. This claiming must be demonstrated and justified. In fact, even if the numbers point to different directions, they respect to different contexts, which is not acknowledge. 

Reviewer suggestion is appreciated. 

The information was incorporated to section 4.1. 

Lines 388 - 412 

Please show more clearly how the section 4.3 relates to the previously presented Results. 

Reviewer suggestion is appreciated. 

The information was incorporated to section 4.3. 

On lines 425-451 

I suggest a new section from line 445, where the merits of this study could be more elaborated.  

Reviewer suggestion is appreciated. 

The information was incorporated to the discussion section as part of the strengths of the study. 

On lines 505-511. 

The disproportion of mathematics students should be included. 

Reviewer suggestion is appreciated. 

Information was incorporated to study limitations on lines 471-476. 

The fact that the data is just from one University and was collected in the COVID-19 period should be addressed. 

Reviewer suggestion is appreciated. The information was incorporated to the discussion section as one of the limitations of this study. 

On lines 466-468 

The “practical implications” (line 456) are not related to the Results of the study, which does not address subjects such as smartphone support of “active methodologies”. 

Reviewer suggestion is appreciated. The information was incorporated into the discussion section as part of the practical implications of the study. 

On lines 491-504 

The Conclusions presented are too general and seem to be more as a short version of results rather than the conclusions of the study. I suggest rewriting the Conclusions section. 

Reviewer suggestion is appreciated. Conclusions section rewritten. 

Finally, the subtitle of the paper puts a question that is not discussed, and I am not convinced that this could be discussed under the subject of this paper. 

Reviewer suggestion is appreciated. The meaning of the subtitle referring to opportunities for Higher Education was addressed in the conclusions. 

  Yours Sincerely, Author´s  

Reviewer 3 Report

Line 11: In the expression "Students connected to the LMS more days in their computers than in their laptops" I think you mean "their smartphones".
Line 38: Remove additional "it".
Lines 214-217: The user_agent gives information about the used device and not the visited material.
Line 228-235: It is not clear what you are doing. What data should have normal distribution? 
Line 268: You collected data in this time period "April 13, 2020, to August 30, 2020", but you have not considered that during Summer learning statistics are different.
Line 280: You inverted "computer" and "smartphones".
Lines 315-316: The number of hours is incorrect.

Actually, it is not clear what you are trying to show. It is not clear if you are trying to demonstrate or to confute your hypothesis. I don't consider so scientifically sounding to demonstrate that students used mostly computers instead of smartphones during pandemic. 

Author Response

Dear, 

We are writing in relation to manuscript ID: education-1304042, entitled " Smartphone use Among Undergraduate STEM Students During COVID-19: An Opportunity for Higher Education?". Authors: Javier Mella-Norambuena, Rubia Cobo-Rendon, Karla Lobos, Fabiola Sáez-Delgado*, Alejandra Maldonado-Trapp.  

We would like to thank for their comments, which we believe have greatly improved the quality of our article. We have modified the text accordingly and respond to each of the issues raised below.  

Table 1. Reviewers’ comments and revisions with corresponding improvement 

Reviewer 3 

Reviewer's comment 

Authors' response 

Line 11: In the expression "Students connected to the LMS more days in their computers than in their laptops" I think you mean "their smartphones". 

Reviewer suggestion is appreciated. The modification was made. 

Line 38: Remove additional "it". 

Reviewer suggestion is appreciated. The modification was made. 

Lines 214-217: The user_agent gives information about the used device and not the visited material 

The paragraph was improved, indicating that the type of resource visited is extracted from the url and the access device from the user_agent 

Line 228-235: It is not clear what you are doing. What data should have normal distribution?  

Reviewer suggestion is appreciated. The modification was made 

Line 268: You collected data in this time period "April 13, 2020, to August 30, 2020", but you have not considered that during Summer learning statistics are different. 

Reviewer suggestion is appreciated. 

The type of activities that take place on the dates presented were claryfied. In Chile, the dates correspond to the activities of the first academic semester. In the region, summer activities are during the month of December. 

Line 280: You inverted "computer" and "smartphones". 

Reviewer suggestion is appreciated. The modification was made. 

Lines 315-316: The number of hours is incorrect. 

Reviewer suggestion is appreciated. The respective modification was made. 

Actually, it is not clear what you are trying to show. It is not clear if you are trying to demonstrate or to confute your hypothesis. I don't consider so scientifically sounding to demonstrate that students used mostly computers instead of smartphones during pandemic.  

Reviewer comment is appreciated. 

It is important to note that the use of smartphones for academic purposes has a great potential. Actually, there are initial descriptive studies that indicate a basis for proposing inferential hypotheses with greater implications for the investigation. In a previous exploration carried out by the authors, the Web of Science Core Collection database was entered using as search algorithm: M-Learning (All Fields) and learning analytics (All Fields), showing 9 results. Of these investigations, only one linked learning analytics with mobile learning. However, no previous evidence was found on the adaptation of LMS’s to M-learning despite its benefits and the fact that smartphones are widely used by students. 

Our general question was, What do students use smartphones for? Thus, we wanted to give guidelines to promote the use of smartphones and generate new learning opportunities in higher education, (see implications under discussion). 

 Yours Sincerely, Author´s  

Round 2

Reviewer 2 Report

I appreciate that the author has embraced the suggestions that were made before, so the article improved accordingly

Reviewer 3 Report

I appreciate you have implemented my comments and have provided answers to my questions.